

# Integrated analysis of lymphocyte infiltration-associated lncRNA for ovarian cancer via TCGA, GTEx and GEO datasets

Meijing Wu[1,*], Xiaobin Shang[2,*], Yue Sun[1], Jing Wu[1] and Guoyan Liu[1]

[1] Department of Gynecology and Obstetrics, Tianjin Medical University General Hospital, Tianjin, China
[2] Department of Esophageal Cancer, Tianjin Medical University Cancer Institute and Hospital, Tianjin, China
[*] These authors contributed equally to this work.

Corresponding author
Guoyan Liu, liuguoyan211@126.com

## ABSTRACT

**Background.** Abnormal expression of long non-coding RNAs (lncRNA) play a significant role in the incidence and progression of high-grade serous ovarian cancer (HGSOC), which is a leading cause of mortality among gynecologic malignant tumor patients. In this study, our aim is to identify lncRNA-associated competing endogenous RNA (ceRNA) axes that could define more reliable prognostic parameters of HGSOC, and to investigate the lncRNAs' potential mechanism of in lymphocyte infiltration.

**Methods.** The RNA-seq and miRNA expression profiles were downloaded from The Cancer Genome Atlas (TCGA) and the Genotype-Tissue Expression (GTEx) database; while for obtaining the differentially expressed lncRNAs (DELs), miRNAs (DEMs), and genes (DEGs), we used edgeR, limma and DESeq2. After validating the RNA, miRNA and gene expressions, using integrated three RNA expression profiles (GSE18520, GSE27651, GSE54388) and miRNA profile (GSE47841) from the Gene Expression Omnibus (GEO) database, we performed Gene Ontology (GO) and Kyoto Encyclopedia of Gene and Genome (KEGG) pathway analyses through ClusterProfiler. The prognostic value of these genes was determined with Kaplan–Meier survival analysis and Cox regression analysis. The ceRNA network was constructed using Cytoscape. The correlation between lncRNAs in ceRNA network and immune infiltrating cells was analyzed by using Tumor IMmune Estimation Resource (TIMER), and gene markers of tumor-infiltrating immune cells were identified using Spearman's correlation after removing the influence of tumor purity.

**Results.** A total of 33 DELs (25 upregulated and eight downregulated), 134 DEMs (76 upregulated and 58 downregulated), and 1,612 DEGs (949 upregulated and 663 downregulated) were detected that could be positively correlated with overall survival (OS) of HGSOC. With the 1,612 analyzed genes, we constructed a ceRNA network, which indicated a pre-dominant involvement of the immune-related pathways. Furthermore, our data revealed that LINC00665 influenced the infiltration level of macrophages and dendritic cells (DCs). On the other hand, FTX and LINC00665, which may play their possible roles through the ceRNA axis, demonstrated a potential to inhibit Tregs and prevent T-cell exhaustion.

**Conclusion.** We defined several prognostic biomarkers for the incidence and progression of HGSOC and constructed a network for ceRNA axes; among which three were indicated to have a positive correlation with lymphocyte infiltration, namely:

FTX-hsa-miR-150-5p-STK11, LINC00665-hsa-miR449b-5p-VAV3 and LINC00665-hsa-miR449b-5p-RRAGD.

## INTRODUCTION

In USA, among the gynecologic malignant tumors, the highest mortality in 2019 was been reported to be ovarian cancer, related with an annual incidence of 22,530 and a mortality of 13,980. This is significantly higher than cervical or endometrial cancers (*Siegel, Miller & Jemal, 2019*). Most deaths (70%) were presented with an advanced-stage, high-grade serous ovarian cancer (HGSOC). The standard treatment encompasses surgery combined with platinum–based chemotherapy. However, most patients died of platinum resistance followed by cancer recurrence (*Liu, Xue & Zhang, 2015*). Therefore, it is of great significance to study the underlying molecular mechanisms of the development of ovarian cancer, which can help us identify more molecular markers for monitoring recurrence, as well as exploring more effective ways to evade drug resistance, and control progression or metastasis.

Accumulating evidence has demonstrated that abnormal expression of long non-coding RNA (lncRNA), which is more specific to tissues and has more quantities in comparison with protein-coding genes (*Cabili et al., 2011*), is involved in the carcinogenesis and progression of ovarian cancer (*Martini et al., 2017*; *Wu et al., 2017*). The lncRNAs can sponge a miRNA through the miRNA response elements (MREs) thereby inhibiting the translation of mRNA, which is propagated as the competing endogenous RNA (ceRNA) hypothesis (*Thomson & Dinger, 2016*). This regulatory mechanism also plays an important role in the development of ovarian cancer. For instance, MALAT1/miR-506/iASPP axis was found to regulate the ovarian cancer growth and DNA synthesis (*Lei et al., 2017*). NEAT1 expression knockdown in platinum resistant (PTX$^r$) ovarian cancers can improve PTX sensitivity, which was mediated by miR-194/ZEB1 axis (*An, Lv & Zhang, 2017*). The overexpression of lncRNA HOTAIR and signaling protein MAPK1 can be reversed by upregulating miR-1, miR-214-3p, or miR-330-5p, indicating a partially attribution to the ceRNA regulatory network (*Yiwei et al., 2015*). However, to date, the studies performed to investigate competing endogenous RNA mechanisms of lncRNAs for HGSOC are rare.

The goal of this investigation was to identify the prognostic value of the lncRNA–miRNA–mRNA interaction axes for elucidating the development of HGSOC, as well as investigating the potential mechanisms of lncRNAs in lymphocyte infiltration. We have identified novel prognostic biomarkers and therapeutic targets for the HGSOC treatment.

## MATERIALS & METHODS

### Data collection and preprocessing

The RNA-seq and miRNA expression profiles of ovarian cancer were downloaded from The Cancer Genome Atlas (TCGA), while the RNA-seq datasets of normal ovarian tissue

were searched from the Genotype-Tissue Expression (GTEx) (*Carithers et al., 2015*). The RNA-seq datasets of counts value from TCGA and GTEx were normalized and processed with TCGAbiolinks package (*Mounir et al., 2019*) (Table S1). It contains RNA expression profile of 363 high grade serous ovarian cancer and 108 normal tissues. The miRNA datasets from TCGA contain microRNA expression profiles of 479 HGSOC and 8 normal tissues (Table S2). On the basis of the information recorded in the HUGO Gene Nomenclature Committee (HGNC; http://www.genenames.org/) (*Eyre et al., 2006*), the lncRNAs and protein-coding genes in RNA expression profiles were annotated. Figure S1 illustrates the workflow for bioinformatics based analysis.

## Differential RNA expression analysis

The differentially expressed protein-coding genes/lncRNAs (DEGs/DELs) from HGSOC and normal ovarian tissues were selected using the edgeR (Version 3.22.5; http://www.bioconductor.org/pack-ages/release/bioc/html/edgeR.html), limma (Version 3.30.0; http://www.bioconductor.org/packages/release/bioc/html/limma.html) and DESeq2 (Version 1.20.0; http://www.bioconductor.org/packages/release/bioc/html/DESeq2.html) package of R software (*Nikolayeva & Robinson, 2014*; *Carvalho et al., 2007*; *Anders & Huber, 2010*). Subsequently, we combined the differentially expressed genes, acquired from the three packages, to get the convergence gene set. On the other hand, differentially expressed miRNAs (DEMs) were identified using the limma package. A false discovery rate (FDR) $lt$ 0.05 and |logFC (fold change)| $>$ 1 were set as the criteria value for DEGs, DEMs and DELs. The expression intensity and direction of DEGs, DELs, and DEMs were represented by heatmaps using the pheatmap R package (Version: 1.0.12; https://cran.r-project.org/web/packages/pheatmap) based on Euclidean distance.

## Validation of DEL, DEM, and DEG expressions

We chose four ovarian cancer datasets from the Gene Expression Omnibus (GEO) database (http://www.ncbi.nlm.nih.gov/geo/). Three of them (GSE18520: 53 vs 10, GSE27651: 22 vs 6, GSE54388: 16 vs 6) were based on the GPL570 platform (Affymetrix Human Genome U133 Plus 2.0 Array) while another one (GSE47841: 12 vs 9) was from GPL14613 platform (Affymetrix Multispecies miRNA-2 Array). In order to improve the number of samples (91 HGSOC samples vs 22 normal samples), we integrated all samples of three datasets (GSE18520, GSE27651, GSE54388), and then removed the batch effect so as to avoid generating less reliable results using sva package (*Leek et al., 2012*) (Table S3). Principal Component Analysis (PCA) was performed on the gene expression profiles obtained from GEO, to reduce the dimensionality and evaluate the independence of normal and tumor samples. The expressions of DELs, DEMs and DEGs were validated in microarray datasets from GEO that detected expression profiles containing mRNA and lncRNA (GSE18520, GSE27651, GSE54388) as well as miRNA (GSE47841) between the HGSOC and normal tissues. The difference of expression was tested by $t$-test. A value of $p < 0.05$ was considered statistically significant.

## Function enrichment analysis

Biological process of Gene Ontology (GO) and Kyoto Encyclopedia of Genes and Genomes (KEGG) analysis of DEGs was conducted by using the clusterProfiler R package (Version 6.8; http://www.bioconductor.org/packages/release/bioc/html/clusterProfiler.html) (*Yu et al., 2012*) to predict their underlying functions. A $p < 0.05$ was considered to be statistically significant.

## Survival analysis

The expressions of DEG/DEL/DEM were extracted and merged with the survival information in each sample (Table S4). Univariate Cox regression analysis was implemented to perform prognostic analysis using the R survival package (Version 2.43.3; https://cran.r-project.org/package=survival). The survival result was expressed as Kaplan–Meier (K-M) curve, with higher expression of a specific gene versus lower expression of this gene in a patient, the cutoff value determined by survminer (Version 0.4.3; https://cran.r-project.org/package=survminer). Again a $p < 0.05$ was considered as related to prognostic significance.

## lncRNA–miRNA–mRNA ceRNA regulatory network construction

The starBase database (http://starbase.sysu.edu.cn/) (*Li et al., 2014*) was used to study the lncRNA-miRNA interactions. The target genes of miRNAs in the DELs–DEMs interaction network were predicted using the miRTarBase database. The negative interaction pairs between DEMs and DEGs/DELs were integrated to construct the DELs–DEMs–DEGs ceRNA network using the Cytoscape software (Version 3.6.1; http://www.cytoscape.org/) (*Kohl, Wiese & Warscheid, 2011*). Furthermore, we analyzed functional enrichment of the genes in the ceRNA network with clusterProfiler R package.

## Immune infiltrates analysis

We analyzed the correlation between lncRNAs in ceRNA network and the abundance of immune cell infiltrates; including B cells, $CD^{4+}$ T cells, $CD^{8+}$ T cells, neutrophils, macrophages, and dendritic cells using Tumor IMmune Estimation Resource (TIMER) (https://cistrome.shinyapps.io/timer/) (*Li et al., 2017*). And the tumor purity was calculated using the estimate R package (https://r-forge.r-project.org/projects/estimate/), In addition, correlations between the lncRNAs and marker genes of tumor-infiltrating lymphocytes were explored using Spearman's correlation. The gene markers of tumor-infiltrating lymphocytes included markers of $CD^{8+}$ T cells, B cells, general T cells, tumor-associated macrophages (TAMs), M1 macrophages, M2 macrophages, monocytes, neutrophils, natural killer (NK) cells, dendritic cells (DC), T helper 1 (Th1) cells, T helper 2 (Th2) cells, T helper 17 (Th17) cells, follicular helper T (Tfh) cells, regulatory T cell (Tregs) and exhausted T cells, which were referenced in prior studies (*Sousa & Maatta, 2016*). The gene expression level is displayed with log2 transformation.
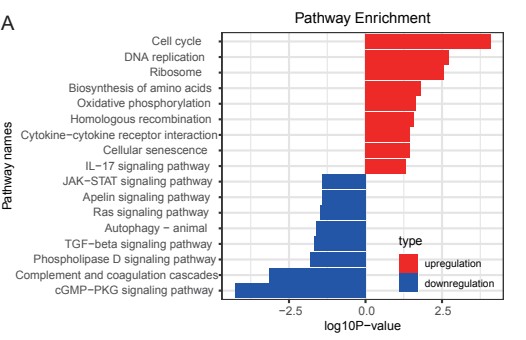
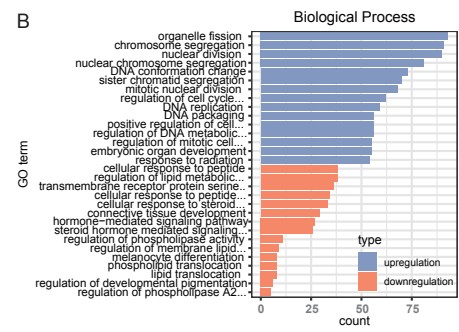

**Figure 1** **Function enrichment analyses for differently expressed genes.** (A) KEGG pathways enrichment for upregulated genes (red) and downregulated genes (blue). (B) Biological process terms of GO enrichment for upregulated genes (blue) and downregulated genes (orange).

## RESULTS

### Differential expression analysis

A total of 629 DELs (343 downregulated and 286 upregulated), 7, 514 DEGs (3, 319 downregulated and 4, 195 upregulated) and 336 DEMs (165 downregulated and 171 upregulated) were identified between HGSOC and normal ovarian tissues based on the threshold of FDR<0.05 and |logFC|>1. The expression changes of DELs (Fig. S2A), DEMs (Fig. S2B), and DEGs (Fig. S2C) were displayed by heatmap analysis.

### Validation of DEL, DEM, and DEG expressions

To validate the expressions of detected differently expressed genes above, we utilized datasets from GEO. The results of PCA indicated that normal samples vs. tumor samples in the merged datasets had a significant difference (Fig. S3). After combining gene expressions in GEO, 111 DELs (77 downregulated and 34 upregulated), 2,620 DEGs (1,535 downregulated and 1,085 upregulated) while 186 DEMs (101 downregulated and 85 upregulated) were identified for subsequent analysis.

### Functional annotation for DEGs

GO and KEGG pathway analysis were performed to explore the potential biological functions of DEGs. The results, through KEGG pathway enrichment analysis, indicated that upregulated genes were mainly involved in cell cycle, oxidative phosphorylation, homologous recombination, cytokine-cytokine receptor interaction and IL-17 signaling pathway, while downregulated genes mainly enriched in cGMP-PKG signaling pathway, complement and coagulation cascades, TGF-β signaling pathway and Ras signaling pathway, as shown in Fig. 1A and Table S5. Furthermore, the GO biological process terms were performed. In line with the KEGG enrichment results, regulation of cell cycle phase transition and positive regulation of cell cycle were obtained for the upregulated genes, while transmembrane receptor protein serine/threonine kinase signaling pathway and regulation of membrane lipid distribution was enriched in the downregulated genes (Fig. 1B and Table S6).

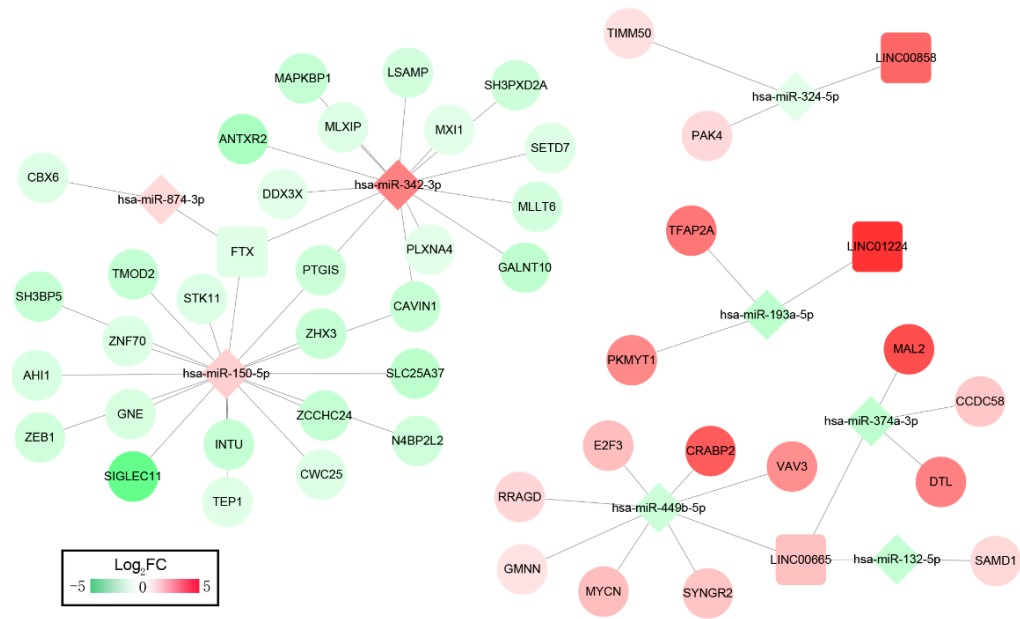

**Figure 2** **ceRNAs interaction network of lncRNA–miRNA–mRNA.** It was consisted of four DELs (three upregulated, one downregulated) and eight DEMs (three upregulated, five downregulated), 44 DEGs (15 upregulated, 29 downregulated). Square nodes represent lncRNAs; diamond nodes represent miRNAs; circular nodes represent mRNAs. The colors present the fold change (higher expression is red, lower expression is green).

## Prognosis prediction for DELs, DEMs, and DEGs

We further performed the survival analysis for DELs, DEMs, and DEGs base on TCGA data and clinical information to excavate the prognostic genes. The results showed that 33 DELs (25 upregulated, 8 downregulated), 134 DEMs (76 upregulated, 58 downregulated), and 1612 DEGs (949 upregulated, 663 downregulated) were significantly associated with overall survival (OS) ($p < 0.05$) (Table S7).

## ceRNA network construction

We then constructed ceRNA networks to explore the interactions between DELs, DEMs and DEGs. The 33 OS-related DELs were mapped into the starBase database to predict their target microRNAs. Comparing with the OS-related DEMs, lncRNA/miRNA interacted networks were constructed. Subsequently, the target genes of selected DEMs were searched using miRTarBase database for identified DEMs–DEGs interaction. After taking the change of expression and prognosis information into consideration, several DELs-DEMs-DEGs interactions including 4 DELs (3 upregulated, 1 downregulated), 8 DEMs (3 upregulated, 5 downregulated) and 44 DEGs (15 upregulated, 29 downregulated) were predicted and resultantly a DELs–DEMs–DEGs ceRNA network was established, as shown in Fig. 2.

Function enrichment analysis showed the genes in the ceRNA network participate in several KEGG pathways; including T cell receptor signaling pathway, MicroRNAs in cancer progression pathway, FoxO signaling pathway, Autophagy-animal and mTOR signaling pathways (Table S8). The DEL–DEM–DEG interactions associated with signaling pathways

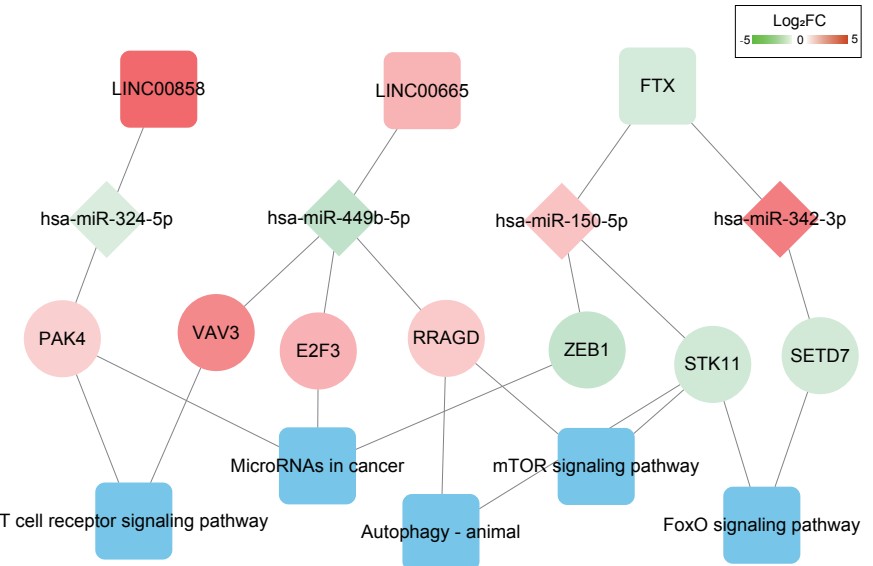

**Figure 3  ceRNAs interaction network of lncRNA–miRNA–mRNA-KEGG pathway.** It was consisted of three DELs (two upregulated, one downregulated), four DEMs (two upregulated, two downregulated), seven DEGs (four upregulated, three downregulated) and five KEGG pathways. Square nodes represent lncRNAs; diamond nodes represent miRNAs; circular nodes represent mRNAs. The colors present the fold change (higher expression is red, lower expression is green). The blue square nodes represent KEGG pathways.

are shown in Fig. 3, which involved 3 DELs (2 upregulated and 1 downregulated), 4 DEMs (2 upregulated and 2 downregulated), and 7 DEGs (4 upregulated and 3 downregulated).

Further K-M curves of genes in the ceRNA network were plotted. The relationships between their respective expressions and the prognostic values were in accordance with our hypothesis. The results showed that patients with a high expression of DELs (LINC00858 and FTX) and DEGs (PAK4, ZEB1, STK11 and SETD7) indicated a poor survival prognosis, meanwhile the high expressions of DEMs (has-miR-324-5p, has-miR-150-5p and has-miR-342-3p) in the ceRNA axes presented and good survival prognosis. Similarly, the high expressions of LINC00665 and the related protein coding genes (VAV3, E2F3, RRAGD) played their roles as suppressor genes, while the high expressions of has-miR-449-5p acted as an oncogene (Fig. 4 and Fig. S4).

## lncRNA expression is correlated with lymphocyte infiltration level in HGSOC

Since the functional analysis indicated that the mRNAs in ceRNA network enriched in immune-related pathways, such as T-cell receptor signaling pathway, Autophagy –animal, and mTOR signaling pathway, whether the lnRNAs played potential roles in lymphocyte infiltration? Tumor-Infiltrating Lymphocyte Grade has been proven to be an independent predictor of sentinel lymph node status, as well as survival in cancers (*Ohtani, 2007*; *Azimi et al., 2012*). However, genomic approaches that are commonly used for the analysis of lymphocyte infiltration in clinical tumor samples can be severely influenced by tumor
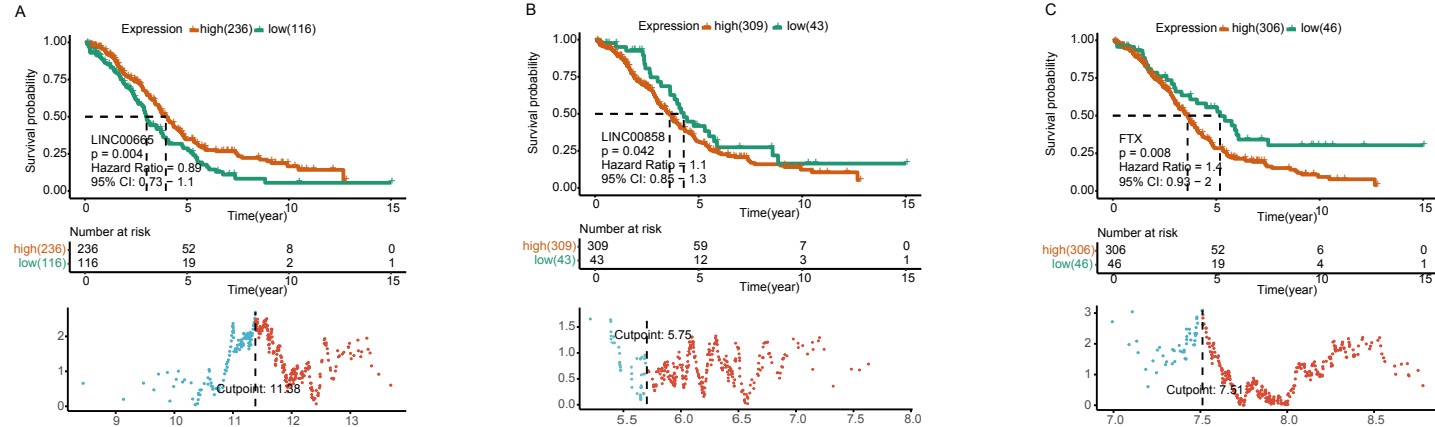

**Figure 4** **Survival analysis of lncRNAs in ceRNA network.** Kaplan–Meier analysis of differentially expressed LINC00665 (A), LINC00858 (B), and FTX (C) in ceRNA network by comparing the higher (red) and lower (green) expressions with overall survival outcomes for patients with HGSOC. *P*-value set for this analysis is less than 0.05. The bottom part shows the threshold set by the R package. Horizontal axis, the expression level of a certain gene; vertical axis, standardized Log-Rank statistic; dotted line, the cutoff value for gene expressions.

purity (*Yoshihara et al., 2013*). We put the expression profiles into TIMER website after eliminating the influence of tumor purity.

The results showed that expression levels of three lncRNAs all have significantly positive correlations with infiltrating levels of tumor purity (Fig. S5, first column). FTX exhibited significant negative correlation with infiltrating levels of $CD^{8+}$ T cells ($r = -0.13$, $P = 0.012$), neutrophils ($r = -0.12$, $P = 0.018$), and macrophages ($r = -0.17$, $P = 0.00091$) as shown in Fig. 5A. LINC00855 also has negative correlations with neutrophils ($r = -0.12$, $P = 0.021$) and DCs ($r = -0.11$, $P = 0.043$) (Fig. 5B). On the other hand, LINC00665 demonstrated significant positive correlation with infiltrating levels of $CD^{4+}$ T cells ($r = 0.11$, $P = 0.044$), negative correlations with $CD^{8+}$ T cells ($r = -0.16$, $P = 0.0026$), neutrophils ($r = -0.27$, $P = 1.8^{e-07}$), macrophages ($r = -0.11$, $P = 0.029$) and DCs ($r = -0.26$, $P = 3.7e-07$) (Fig. 5C). These findings suggest that three lncRNAs, LINC00855, FTX and LINC00665, play a specific role in lymphocyte infiltration in HGSOC, especially LINC00665.

## Correlation analysis between lncRNA expression and immune marker sets

We further investigated the correlations between lncRNAs and marker sets of different immune cells, including $CD^{8+}$ T cells, T cells (general), B cells, monocytes, TAMs, M1 and M2 macrophages, neutrophils, NK cells, and DCs. Different type of T cells were also analyzed, which include Th1 cells, Th2 cells, Tfh cells, Th17 cells, Tregs, as well as exhausted T cells (Table 1 and Table S9).

After adjusting correlation by purity, the results revealed that most immune markers of various immune cells and T cells are significantly correlated with the FTX and LINC00665 expression level. However, only 22 gene markers were found to be correlated with the expression level of LINC00885.

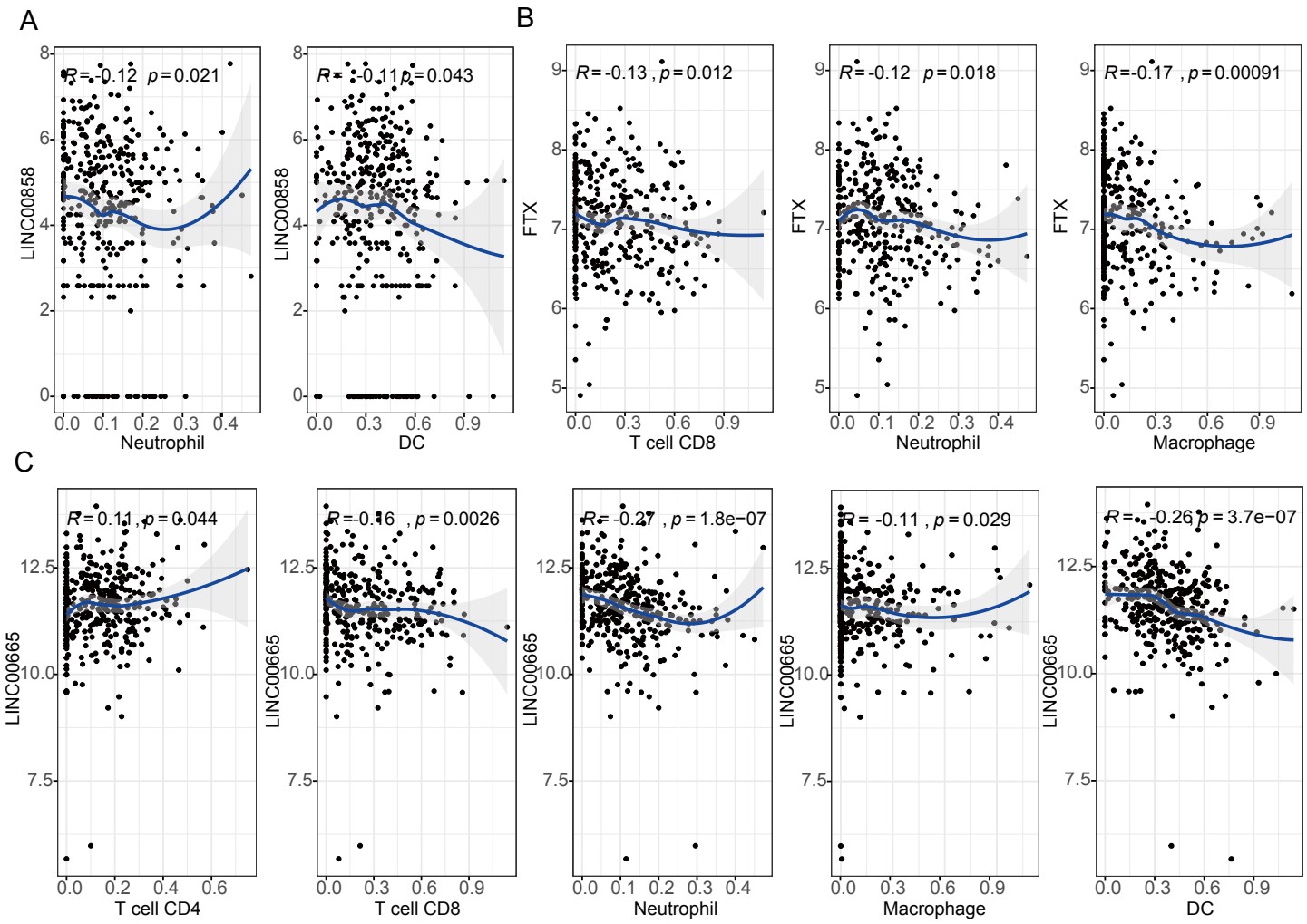

**Figure 5  Correlation of lncRNAs expression with lymphocyte infiltration level in high-grade serous ovarian cancer.** (A) LINC00855 has negative correlations with neutrophils and DCs. (B) FTX has negative correlations with infiltrating levels of CD8+ T cells, neutrophils, and macrophages. (C) LINC00665 has significant positive correlations with infiltrating levels of CD4+ T cells, negative correlations with CD8+ T cells, neutrophils, macrophages and DCs.

It is also found that the expression levels of most markers of monocytes, TAMs, M2 macrophages have strong correlations with LINC00665 expression.

As is shown in Fig. 5, high LINC00665 expression correlates with low infiltration level of DCs. This result can be explained by the following findings that DC markers such as HLA-DPB1, HLA-DQB1, HLA-DRA, HLA-DPA1, CD1C (BDCA-1) and ITGAX (CD11c) show significant negative correlations with LINC00665 expression. We can further conclude that there is a significant negative correlation between LINC00665 and DC infiltration. This finding is rather interesting because tumor metastasis can be aggravated by DC reducing CD$^{8+}$ T cell cytotoxicity and increasing Treg cells (*Sawant et al., 2012*).

It is also found that there are significant correlations between FTX, LINC00665 and marker genes of Treg and T cell exhaustion, such as FOXP3, TGFB1 (TGFβ), IL10, TBX21

Table 1 Correlation analysis between lncRNAs and related genes and markers of immune cells after eliminating the influence of tumor purity in HGSOC.

| | | LINC00858 | | FTX | | LINC00665 | |
|---|---|---|---|---|---|---|---|
| Description | Gene markers | Cor | P | Cor | P | Cor | P |
| CD8+ T cell | CD8A | −0.095 | 0.071 | −0.095 | 0.07 | −0.204 | *** |
| | CD8B | 0.088 | 0.092 | −0.15 | * | −0.078 | 0.14 |
| B cell | CD19 | −0.016 | 0.762 | −0.018 | 0.738 | 0.07 | 0.186 |
| | CD79A | −0.053 | 0.317 | −0.126 | 0.017 | −0.05 | 0.341 |
| TAM | CCL2 | −0.166 | * | −0.117 | 0.025 | −0.197 | ** |
| | CD68 | −0.182 | ** | −0.209 | *** | −0.298 | *** |
| | IL10 | −0.072 | 0.172 | −0.092 | 0.081 | −0.236 | *** |
| M1 Macrophage | NOS2(INOS) | 0.002 | 0.965 | −0.093 | 0.077 | −0.08 | 0.13 |
| | IRF5 | −0.059 | 0.262 | −0.201 | ** | −0.123 | 0.019 |
| | PTGS2(COX2) | −0.044 | 0.4 | −0.078 | 0.137 | −0.068 | 0.198 |
| M2 Macrophage | CD163 | −0.202 | ** | −0.153 | * | −0.252 | *** |
| | VSIG4 | −0.136 | * | −0.126 | 0.016 | −0.271 | *** |
| | MS4A4A | −0.147 | * | −0.187 | ** | −0.288 | *** |
| Dendritic cell | HLA-DPB1 | −0.083 | 0.115 | 0 | 0.997 | −0.224 | *** |
| | HLA-DQB1 | −0.059 | 0.263 | 0.056 | 0.284 | −0.185 | ** |
| | HLA-DRA | −0.102 | 0.052 | −0.02 | 0.71 | −0.213 | *** |
| | HLA-DPA1 | −0.089 | 0.089 | −0.008 | 0.872 | −0.234 | *** |
| | CD1C(BDCA-1) | −0.204 | *** | −0.011 | 0.827 | −0.268 | *** |
| | NRP1(BDCA-4) | −0.107 | 0.041 | −0.161 | * | −0.111 | 0.034 |
| | ITGAX(CD11c) | −0.165 | * | −0.066 | 0.21 | −0.239 | *** |
| Th1 | TBX21(T-bet) | −0.183 | ** | −0.176 | ** | −0.206 | *** |
| | STAT4 | −0.169 | * | −0.143 | * | −0.238 | *** |
| | IFNG(IFN-$\gamma$) | −0.05 | 0.346 | −0.103 | 0.051 | −0.128 | 0.014 |
| | CXCR3 | −0.112 | 0.033 | −0.143 | * | −0.222 | *** |
| | CCR5 | −0.144 | * | −0.146 | * | −0.279 | *** |
| Th2 | GATA3 | −0.128 | 0.015 | −0.061 | 0.244 | −0.212 | *** |
| | STAT6 | −0.013 | 0.811 | 0.16 | * | −0.097 | 0.064 |
| | CCR4 | −0.133 | 0.011 | −0.114 | 0.03 | −0.245 | *** |
| Th17 | STAT3 | −0.067 | 0.203 | −0.227 | *** | −0.236 | *** |
| | IL6 | −0.028 | 0.589 | −0.171 | * | −0.142 | * |
| | CCR6 | −0.134 | 0.01 | −0.05 | 0.344 | −0.207 | *** |
| | RORA(ROR$\alpha$) | 0.041 | 0.44 | 0.129 | 0.014 | −0.144 | * |
| tTreg | FOXP3 | 0.002 | 0.964 | −0.192 | ** | −0.19 | ** |
| | TGFB1(TGF$\beta$) | −0.133 | 0.012 | −0.229 | *** | −0.091 | 0.084 |
| | IL10 | −0.072 | 0.172 | −0.092 | 0.081 | −0.236 | *** |
| | TBX21(T-bet) | −0.183 | ** | −0.176 | ** | −0.206 | *** |
| T cell exhaustion | PDCD1(PD-1) | −0.034 | 0.522 | −0.193 | ** | −0.173 | ** |
| | CTLA4 | −0.11 | 0.037 | −0.19 | ** | −0.234 | *** |
| | LAG3 | −0.096 | 0.066 | −0.135 | * | −0.091 | 0.083 |
| | HAVCR2(TIM-3) | −0.134 | 0.011 | −0.128 | * | −0.297 | *** |

**Table 1** (*continued*)

| Description | Gene markers | LINC00858 | | FTX | | LINC00665 | |
|---|---|---|---|---|---|---|---|
| | | Cor | P | Cor | P | Cor | P |
| | GZMB | −0.085 | 0.107 | −0.108 | 0.04 | −0.169 | * |

**Notes.**

lncRNA, long-non-coding RNA; HGSOC, high-grade serous ovarian cancer; TAM, tumor-associated macrophage; Th, T helper cell; Treg, regulatory T cell; Cor, R value of Spearman's correlation.

Level of significance: $^{*}p < 0.01$. $^{**}p < 0.001$. $^{***}p < 0.0001$.

(T-bet), PDCD1 (PD-1), CTLA4 and HAVCR2 (TIM-3). FOXP3 is a critical marker of Treg cells which suppress cytotoxic T cells attacking tumor cells (*Facciabene, Motz & Coukos, 2012*). We found that FTX and LINC00665 expressions have strong negative correlations with TIM-3, which is known as a critical factor that regulates T cell exhaustion. This finding suggests that these two lncRNA expressions are of great importance in TIM-3 mediating T cell exhaustion (Table 1). Accordingly, it is further confirmed that FTX and LINC00665 are specifically correlated with immune infiltrating cells, suggesting that these two lncRNAs play vital roles in inhibiting immune escape in the ovarian microenvironment.

# DISCUSSION

It is increasingly clear that abnormal lncRNA expression might contribute to tumorigenesis and progression of high-grade serous ovarian cancer (HGSOC). The ceRNA hypothesis, in which lncRNAs play their role at the post-transcriptional level, has been considered as a novel measure of gene regulation. However, comprehensive analyses of differentially expressed profiles of lncRNAs, miRNAs and mRNA, as well as their interactions through integrating TCGA, GTEx and GEO datasets have not yet been reported.

In the present study, we obtained complete datasets of lncRNA, microRNA and mRNA through differential gene expression analysis after analyzing datasets from TCGA and GTEx and integrated validation datasets from GEO. And then these genes were submitted for survival analysis for identifying OS-associated DELs, DEMs and DEGs. We further developed a prognostic ceRNA network with the detected OS-associated genes, which contain 33 DELs (25 upregulated, 8 downregulated), 134 DEMs (76 upregulated, 58 downregulated) and 1, 612 DEGs (949 upregulated, 663 downregulated).

The ceRNA network was constructed by using four lncRNAs, LINC01224, FTX, LINC00855 and LINC00665. Considering that mRNAs are the implementers of molecular function, KEGG analysis were performed to annotate the functions of DEGs influenced by the four lncRNAs in the ceRNA network. Three lncRNAs, LINC00665, LINC00855 and FTX, were left with their target genes involved in several pathways, one of which was miRNA in cancer pathway, indicating that they may act partly in a ceRNA network to exert their functional role.

To get a better understanding of the molecular functions of these three OS-associated lncRNAs, we reviewed the published researches of the same lncRNAs. LINC00665 is upregulated in HGSOC, it is also overexpressed in hepatocellular carcinoma (HCC) and lung cancer patients, but act as an oncogene with poor prognosis (*Wen et al., 2018*), which is contrary to our univariate Cox regression analysis (HR:0.89, 95% CI [0.73–1.1]). In lung

adenocarcinoma (LUAD) tissues, it played its roles through linc00665-miR98-AKR1B10 axis (*Cong et al., 2019*).

LINC00858 expression was observed to be up-regulated in colorectal cancer (CRC), osteosarcoma and non-small cell lung cancer (NSCLC) tissues, and associated with poor prognosis, which is in line with our hypothesis, and it exerted its functional effect through the miR-22-3p/YWHAZ axis in CRC (*Sha et al., 2019*). thus promoting tumorigenesis through LINC00858/miR-139/CDK14 axis in osteosarcoma (*Gu et al., 2018*), as a ceRNA for miR-422a to facilitate cell proliferation in NSCLC (*Zhu et al., 2017*).

The long non-coding RNA FTX is an X-inactive-specific transcript (XIST) regulator transcribed from the X chromosome inactivation center. Similar to our results, a significant decrease of FTX in HCC tissues was observed, and patients with higher FTX expression exhibited longer survival, it acted as a tumor suppressor through binding with miR-374a and MCM2 (*Liu et al., 2016*). While in another research, Li et al. reported that FTX was upregulated in HCC, acting as a promoter of tumor progression (*Li et al., 2018*). And it was also upregulated in CRC, gliomas, renal cell carcinoma (RCC) tissue and indicated poor prognosis of patients with CRC (*Yang et al., 2018*) and RCC (*He et al., 2017*), it bound miR-215 and vimentin in CRC, negatively regulating miR-342-3p in gliomas to promote proliferation and invasion (*Zhang et al., 2017*). miR-342-3p was also predicted to interact with FTX in our result.

These results indicate that LINC00665, LINC00855 and FTX can perform their function through miRNA competition as a part of ceRNA axes. But more experiments are needed to validate the roles of these ceRNA axes in HGSOC.

The tumorigenesis and progression not only ascribe to gene variation, but also have strong connection with tumor microenvironment. The tumor-infiltrating lymphocytes (TILs), as components of ovarian cancer tumor microenvironment, have already been shown to have correlation with OS. A previous study divided HGSOC into four molecular subtypes namely, ''immunoreactive,'' ''differentiated,'' ''proliferative,'' and ''mesenchymal'' on the basis of gene expressions obtained from TCGA. And it is found that the prognosis of immunoreactive subtype is better than the other subtypes (*Konecny et al., 2014*), indicating the immune factors are of vital importance in the development of HGSOC.

Interestingly, the KEGG analysis showed that DELs and their target DEGs were associated with immune-related pathway, such as T cell receptor signaling pathway, Autophagy and mTOR signaling pathway, suggesting the correlations between the DELs and the activation of the immune system.

Taking advantage of the website TIMER and co-expression analysis, we found that there were negative correlations between LINC00665 expression level and infiltration level of DCs and macrophages. Moreover, the relationships between LINC00665 expression and the gene markers of immune cells supported the result of lymphocyte infiltration. First, gene markers of M1 macrophages such as NOS2 (INOS), IRF5 and PTGS2 (COX2) were weakly correlated with LINC00665 expression, whereas markers of M2 macrophage such as CD163, MS4A4A and VSIG4 showed moderate and strong correlations. These results revealed that LINC00665 may regulate the tumor-associated macrophage (TAM) polarization.

Our results also demonstrated that FTX and LINC00665 had the potential to inhibit Tregs and prevent T cell exhaustion. The decreased FTX and increased LINC00665 expression negatively correlated with the expression of Treg and T cell exhaustion markers (FOXP3, TGFB1 (TGFβ), IL10, TBX21 (T-bet), PDCD1 (PD-1), CTLA4 and HAVCR2 (TIM-3)).

FTX and LINC00665 also regulated several marker genes of T helper cells (Th1, Th2, Th17 and Tfh). These correlations could reveal a potential mechanism where they regulate T cell functions in HGSOC. Together, these findings suggested that FTX and LINC00665, especially LINC00665, play important roles in regulation of immune infiltrating cells in HGSOC.

Our result defined several ceRNA axes that may influence the tumor immune microenvironment. FTX-hsa-miR-150-5p-STK11 was associated with mTOR signaling pathway and autophagy. STK11 (LKB1) inhibited mTOR signaling. STK11 silencing inhibited autophagy induction and tumor inhibition (*Chung et al., 2017*). Akt/mTOR signaling pathway is a well-known pathway which negatively regulates autophagy (*Hanahan & Weinberg, 2000*). mTOR is an evolutionarily conserved serine-threonine kinase, which has emerged as a central regulator of T-cell lineage specification (*Delgoffe et al., 2011*). In particular, constitutively active mTOR and Akt caused abrogation of Treg differentiation (*Haxhinasto, Mathis & Benoist, 2008*).

LINC00665-hsa-miR449b-5p-RRAGD was associated with mTOR signaling pathway and autophagy as well. RRAGD promoted mTORC1 activity and tumor growth (*Di Malta et al., 2017*), influencing the T-cell similarly. LINC00665-hsa-miR449b-5p-VAV3 was relative with T cell receptor signaling pathway. Researchers observed signal-induced tyrosine phosphorylation of VAV1 and VAV3, which are cytosolic signaling scaffolds and guanine nucleotide exchange factors that can play context-specific roles in immune receptor pathways (*Bustelo, 2014*).

# CONCLUSIONS

In summary, we defined two immune-associated lncRNAs, FTX and LINC00665, which can act as prognostic biomarkers in HGSOC, and constructed three ceRNA axes: FTX-hsa-miR-150-5p-STK11, LINC00665-hsa-miR449b-5p-VAV3 and LINC00665-hsa-miR449b-5p-RRAGD.

## Funding

This work was supported by the National Natural Science Foundation of China (No. 81472761). The funders had no role in study design, data collection and analysis, decision to publish, or preparation of the manuscript.

## Grant Disclosures

The following grant information was disclosed by the authors:
National Natural Science Foundation of China: 81472761.

## Competing Interests

The authors declare there are no competing interests.

## Author Contributions

- Meijing Wu and Xiaobin Shang conceived and designed the experiments, analyzed the data, prepared figures and/or tables, and approved the final draft.
- Yue Sun and Jing Wu performed the experiments, authored or reviewed drafts of the paper, and approved the final draft.
- Guoyan Liu conceived and designed the experiments, authored or reviewed drafts of the paper, and approved the final draft.

## Data Availability

Raw data is available in the Supplementary Files.

## Supplemental Information

Supplemental information for this article can be found online at http://dx.doi.org/10.7717/peerj.8961#supplemental-information.

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
