# Peer review of "Integrated analysis of lymphocyte infiltration-associated lncRNA for ovarian cancer via TCGA, GTEx and GEO datasets"

_PeerJ, doi:10.7717/peerj.8961_

## Round 0.1 · original submission · Major Revisions

Three experts in the field evaluated this submission. They all have concerns related to this manuscript. In my view, the paper needs a major revision.

Reviewer 1 ·

Basic reporting

As for the writing style, I recommend re-writing or re-organizing the 'results' section of the paper. In most of the paragraphs in results section, two major components are missing: rationale to conduct the experiment, and brief summary of the meaning of results. As the writing stands so far, the 'results' section is much too descriptive and contain some methodology. I suggest only showing key numbers and methods that are relevant to understanding the key findings of each section, and additional details can be relocated to the methods section or supplementary materials.

Figures and tables are cluttered with information, which prevents the main points from standing out. Therefore, I recommend condensing figures and tables, and moving some of them to supplementary materials. For example, Figure 1 can be moved to supplementary materials. Table 1 can either be moved to supplementary materials or extracted for the necessary rows/columns for the readers to understand the point being made. Figure 5 and 6 do not have to present each of these experiments - representative examples should be selected, shown, and discussed to make the key points, and other panels should be moved to supplementary materials. Figure 5 is complicated, but it was only mentioned once in the text without any mentioning of what the patterns of the figures mean.

Experimental design

The experiment performed in this paper meets the aims and scope of the PeerJ. The goal of trying to find lncRNA-miRNA-mRNA interaction axes have been well laid out in intro, although emphasizing these in results and how the overarching question is linked to each experiment will strengthen the paper. Additionally, the paper will be much stronger if the authors can provide the background of why using lncRNA as a biomarker of cancer is optimal compared to other existing techniques.

As stated above, the article will be much stronger if the authors can guide the readers through the logic in the results section of the paper. I recommend (1) add a sentence to the beginning of each paragraph laying out the reason to conduct these experiments and the hypothesis being tested or open-ended questions being asked; (2) briefly summarize the findings at the end of each paragraph.

Validity of the findings

The correlation between lncRNA expression and lymphocyte infiltration is weak. Certain panels of Figure 6 clearly show high p-value and lack of correlation. The authors focus on the ones that are correlated, but this type of 'cherry-picking' can only be done if there is clear biological meaning behind the specific pair of comparison. Otherwise, a clear reason of the pairs that do not show correlation should be given. The p-values of Figure 6 are also subject to multiple testing correction because all lncRNAs were tested against all lymphocyte infiltration markers trying to draw a conclusion of their overall correlation.

Additional comments

In this manuscript, Wu et al. profile lncRNA and miRNA in high-grade serious ovarian cancer. The authors propose several lncRNA-miRNA-mRNA axes as biomarkers of this cancer. I appreciate the large amount of work in this study including data collection and detailed analysis, but I would recommend a revision to let the main message of the paper stand out more clearly to the readers.

Reviewer 2 ·

Basic reporting

The article is written in an unqualified format, including alignment, cited location of references (Usually before periods), the use of spaces and punctuations, and the mixture of abbreviation and full name. Writing (logic and language) needs to be improved.

Experimental design

The RNA-seq datasets used in the study were downloaded from TCGA and GTEx. What is the specific form of the data (fpkm, Rpkm or TPM)? The authors need to specify this form in detail. In addition, the batch effects also need to be considered.
I am afraid these will effect the following differential analysis results (The number of differentially expressed genes is so large).
In line 153, the gene expression level is displayed with log2 transformation. The authors did not explain why this process is needed.
The validation data sets used in this paper are only used for differentially expressed analysis. Further analysis on the validation data and results comparing are not implemented.

Validity of the findings

The correlation between lncRNAs and marker genes of tumor-infiltrating lymphocytes, and the correlation between the ceRNA biomarkers and lymphocyte infiltration are very weak indeed (<0.3). So the reliability of the correlation analysis results is questionable.

Additional comments

Differentially expressed lncRNAs, miRNAs, and genes are detected that could be
positively correlated with overall survival of HGSOC. In addition , they constructed a ceRNA network, which indicated a pre-dominant involvement of the immune-related
pathways. They defined several prognostic biomarkers for the incidence and progression of HGSOC and constructed a network for ceRNA axes; among which three have a positive correlation with lymphocyte infiltration, namely; FTX-hsa-miR-150-5p-STK11, LINC00665-hsa-miR449b-5p-VAV3 and LINC00665-hsa-miR449b-5p-RRAGD.

Reviewer 3 ·

Basic reporting

The paper focuses on a subject of interest with a design that seems correct. Reusing available data is an important area of current research.
The article is written in a correct English and is unambiguous. The state-of-the-art is sufficient and it is easy to follow what had been done.
Unfortunately, the Figures are too often difficult to read (too small and overwritten, especially Figures 2, 5 and 6), this greatly diminishes the interest of reading.

Experimental design

Research question is well defined, relevant and meaningful. The design is coherent and the analysis in depth.
On the other hand, it lacks a little critical sense. The main point is the discussion of the conservation of different genes, experiences, and the impact of these choices on the final list. This point is not discussed enough and remains critical and objectionable.
Methods are properly described with sufficient information to be reproducible. This is a very good point.

Validity of the findings

Of course, it would be easy to say (i) the data are not generated locally and above all (ii) no experimental confirmation has been made (translation = from in silico).
I would say that (i) it is therefore a good thing, we are in the era of big data and the use of available data and (ii) it is perfect it allows proposing new hypotheses that are not sometimes not even possible to test at the moment.
My critical point is the description of the choices that leads precisely to the list, the article is too descriptive and lacks criticism on these points.

Additional comments

an interesting approach, a well-conducted article, which needs a little improvement to make it excellent.

---

## Round 0.2 · accepted · Accept

The authors carried out all modifications indicated by the reviewers. In my view, the revised version of the manuscript improved and can be accepted for publication as it is.

Reviewer 1 ·

Basic reporting

See "general comments".

Experimental design

See "general comments".

Validity of the findings

See "general comments".

Additional comments

Most of the major issues have been addressed in the revision. I recommend accepting the manuscript.

Reviewer 3 ·

Basic reporting

The authors answered most of my questions.
The manuscript has been greatly improved.

Experimental design

The authors answered most of my questions.
The manuscript has been greatly improved.

Validity of the findings

The authors answered most of my questions.
The manuscript has been greatly improved.

Additional comments

The authors answered most of my questions.
The manuscript has been greatly improved.